# Phytoplankton Size Structure and Diversity in the Transitional System of the Aquatina Lagoon (Southern Adriatic Sea, Mediterranean)

**DOI:** 10.3390/microorganisms11051277

**Published:** 2023-05-13

**Authors:** Carmela Caroppo, Maurizio Pinna, Maria Rosaria Vadrucci

**Affiliations:** 1National Research Council, Water Research Institute (CNR-IRSA), 74123 Taranto, Italy; 2NBFC—National Biodiversity Future Center, 90133 Palermo, Italy; maurizio.pinna@unisalento.it; 3Department of Biological and Environmental Sciences and Technologies, DiSTeBA, University of Salento, Via Monteroni 165, 73100 Lecce, Italy; 4Research Centre for Fisheries and Aquaculture of Aquatina di Frigole, DiSTeBA, University of Salento, 73100 Lecce, Italy; 5Agenzia Regionale Protezione Ambiente della Campania, 83100 Avellino, Italy; mrvadrucci@gmail.com

**Keywords:** distribution, dynamics, environmental variables, hydrology, size–structure, transitional water ecosystems

## Abstract

The Aquatina Lagoon (Southern Adriatic Sea, Mediterranean Ecoregion) is a transitional water ecosystem with great ecological and socio-economic interest. Anthropogenic activities around the lagoon (e.g., agriculture and tourism) and hydrology can affect the environmental quality and biodiversity of the lagoon. Herein, the dynamics and diversity of phytoplankton communities were studied before and after the opening of a new canal connecting the lagoon with the sea, by using different approaches based on an evaluation of the size and structure of the phytoplankton as well as the taxonomic analyses. The lagoon depicted time-related fluctuations in chemical-physical parameters. The phytoplankton trend was characterized by an increase in abundance and biomass in summer, when pico-sized autotrophs dominated. Generally, nano-sized phytoflagellates dominated the community, while micro-sized dinoflagellates and diatoms were less abundant. An increase in the phytoplankton taxa number was observed throughout the years. All the analyzed parameters were generally relatively homogeneous before the opening of the channel, while some quantitative differences among stations were observed in the second sampling period. Considering the statistical evidence, both environmental and biological parameters were affected by the “dilution” effect exerted by marine water inputs. This research supports the evidence that phytoplankton is a good indicator of the environmental status, and the obtained results contribute to the implementation of management strategies for the conservation of transitional water ecosystems.

## 1. Introduction

A specific ecological characteristic of coastal lagoons is their intrinsic habitat heterogeneity, which is well-documented on the spatial and temporal scales [1,2,3,4]. For this peculiarity, these ecosystems are highly variable and particularly vulnerable to changes on the long temporal scale due to global warming [5,6,7], or as a consequence of changes in geomorphological and hydrological features due to anthropogenic activities over time.

Phytoplankton of transitional water ecosystems include a wide variety of prokaryotic and eukaryotic photosynthetic organisms belonging to various taxonomic groups, such as Cyanophyta, Bacillariophyta, Dinophyta, Chlorophyta, Euglenophyta, Chrysophyta, Cryptophyta, and Xanthophyta [8]. Studies on phytoplankton’s response to the environmental heterogeneity of coastal lagoons have demonstrated the direct effects of abiotic, hydrological, and geomorphological factors on biodiversity and the spatio-temporal dynamics of the phytoplankton community structure [4,9,10,11]. These studies analyzed patterns of variation of phytoplankton at taxonomic as well as morphological and functional levels, and demonstrated that phytoplankton’s growth is controlled by a combination of the temperature and the availability of light and nutrients [12]. The latter (e.g., nitrogen, phosphate) are, in turn, controlled by physical processes such as water exchange with the sea and input from freshwaters, water circulation, water residence time, and seasonal light availability [13]. Despite the fact that the mechanisms that regulate the biodiversity of coastal lagoons are partly explained, much of taxonomic biodiversity remains unknown. This is due to the difficulty of analyzing phytoplankton samples with traditional microscopic techniques, since the samples are often rich in detritus. Furthermore, the presence of small phytoplankton species and/or species that undergo changes in their morphology due to the effect of fixatives makes their identification with optical microscopy difficult [14]. After all, in coastal lagoons, phytoplankton communities are heterogeneous, often represented by a mixture of taxa typically of both the marine and freshwater ecosystems [15].

The Aquatina Lagoon (Southern Adriatic Sea, Mediterranean Ecoregion) is a transitional water ecosystem, in particular a “non-tidal coastal lagoon”, recently included in the Coastal Research Centre of the University of Salento. It was realized firstly as the Research Centre for Experimental Aquaculture, with the aim of improving the research activities applied to aquaculture and fishing. In relation to its nutrient levels, the lagoon is classified as meso-oligotrophic. The muddy seabed is often covered by meadows of *Cymodocea nodosa* and *Ruppia* sp. Moreover, this site hosts a highly taxonomic diversity of terrestrial and aquatic species (plants, algae, zooplankton, macroinvertebrate assemblages, fishes, amphibians, reptiles, birds, and mammals) [16]. The lagoon displays high vulnerability with respect to the human activities carried out in the catchment area of the basin, where agriculture is responsible for the production of around 300 tons and 3 tons per year of fertilizers and pesticides, respectively [17].

The Aquatina Lagoon is included in the European NATURA 2000 Network, according to the Habitat Directive 92/43/CEE: “Aquatina Frigole” (Cod. IT9150003) for the presence of the habitat 1150* (coastal lagoons) and of a *Posidonia oceanica* meadow in the marine area included in the NATURA 2000 site perimeter; for these reasons, it requires specific measures of conservation. Moreover, the Aquatina Lagoon belongs to both LTER-Italy, the national Long-Term Ecological Research network, and the larger European network eLTER [18,19].

Besides the importance of this lagoon, the research on the phytoplankton community is scarce. The phytoplankton community’s structure and the related abiotic factors of the Aquatina Lagoon have been studied since 1985, but not continuously. The first research was carried out by Tolomio et al. [20], followed by Vadrucci et al. [21,22] and Caroppo [23]. These data evidenced irregular patterns across the years, without any evident temporal trend [19]. In 2004, this coastal water system underwent an important modification for the improvement of the water exchanges with the sea: the opening and extension of new canals connecting the lagoon with sea. Due to the relevance of the physical alteration, it is of particular interest to the study of phytoplankton’s dynamics and composition after changes in hydrological features. In fact, studying phytoplankton has important significance for environmental management, since it is the only micro-planktonic indicator of the water quality in transitional water ecosystems, according to the European Water Framework Directive [24] and as has been consequently evidenced in much research [25].

This paper aims to evaluate the phytoplankton dynamics and diversity in two hydrological conditions (before and after the opening of a new canal connecting the lagoon with the sea) by using different approaches based on the evaluation of the phytoplankton’s size and composition as well as of the taxonomic analyses. Our study makes a valuable contribution to understanding how a hydrological regime can influence phytoplankton communities in similar environments, and could be useful in developing action plans to support the conservation of the Aquatina Lagoon and its diversity.

## 2. Materials and Methods

### 2.1. Study Area

The Aquatina Lagoon is an artificial brackish-water basin located on the Adriatic Sea shoreline of the Salento Peninsula (40°27′22″ N–18°12′24″ E) since the 1920s [26]. It exhibits all ecological features of Mediterranean coastal lagoons [27]. It is included within the xero-Mediterranean coastal system [28]. The lagoon exhibits a Y-shaped morphology; is about 2 km long, with a surface area of about 43 ha; and has a mean depth of about 1.2 m. It is connected to the sea by a channel, which is 15 m wide and 400 m long, located in the southernmost area (Figure 1). The principal freshwater inputs are a lateral branch of the Giammatteo Canal (on the northern boundary of the lake), a limited agricultural drainage network, and rainfall. At its NW extremity, a branch of the Giammatteo Canal runs into the lagoon; at its SE extremity, the lake is directly connected to the Adriatic Sea. A third canal, ca. 400 m long and parallel to the coastline, was excavated to connect the SE extremity of the basin with the sea. Nowadays, a central muddy canal and the original southern mouth, opened again in 2004, are present [29]. A positive salinity gradient extends from the northern to the southern area of the lagoon. The mean annual water residence time is 3 days (minimum in winter, 2 days; maximum in summer, 8 days) and the lagoon exports water to the sea during the entire year (42,000 m^3^ d^−1^), with minimum values in the summer (8700 m^3^ d^−1^) and maximum ones in the winter (76,700 m^3^ d^−1^) [30]. Aquatina is a non-tidal lagoon, and the tidal regime, on an annual basis, does not usually exceed 40 cm.

From January to December 1996 and from April 2007 to March 2008, monthly samplings were conducted at three stations, representative of different hydro-biological conditions. Station 1 was located near the connection channel with the Adriatic Sea, Station 2 near the Giammatteo Canal, and Station 3 in the closed branch of the basin (Figure 1).

### 2.2. Sample Collection and Abiotic Factors

Water samples were taken at the surface of each station (*n* = 3) using a 5-L Niskin bottle. Transparency was measured by a Secchi disk. Temperature, salinity, and dissolved oxygen were recorded by an Idronaut Ocean Seven 501 multiprobe and compared with in situ (electronic thermometers) and laboratory (Guildline Autosal 8400 B salinometer and Winkler method) measurements. Nutrient concentrations (N-NH_4_^+^, N-NO_2_^–^, N-NO_3_^–^, P-PO_4_^3–^) were evaluated using spectrophotometry, as described in the methods of Strickland and Parsons [31]. Using Whatman GF/F filters, 500 mL of water were filtered, and sub-samples (50 mL) were used for the analysis of each nutrient. N-NH_4_^+^ was determined by measuring the absorbance at 630 nm of the indophenol complex formed by ammonium in presence of sodium nitroprusside after oxidation with hypochlorite and phenol in an alkaline citrate solution. N-NO_2_^-^ and N-NO_3_^−^ were analyzed using the same spectrophotometric method as that used for the previous reduction of N-NO_3_^−^ in N-NO_2_^−^, which was carried out after exposure of the sample to copper-coated cadmium. The determination of N-NO_2_^−^ + N-NO_3_^−^ involved the formation of a diazo compound by nitrite and sulfanilamide in acidic solution and the subsequent production of a diazo dye in the presence of N-(1-Naphthyl) ethylenediamine dihydrochloride. The absorbance of diazo dye was measured at 543 nm. P-PO_4_^3−^ was evaluated after the production of a blue-colored phosphomolybdic complex derived from soluble reactive phosphate, molybdic acid, ascorbic acid, and trivalent antimony. The absorbance of the colored complex was determined at 885 nm. A Varian Cary 50 UV/Vis spectrophotometer (Varian Inc., Middelburg, The Netherlands) was used for all the measurements. To increase the precision of the analyses, 10 cm optical glass cuvettes were used. The nutrient concentrations were calculated on the bases of the concentration–absorbance curves, which were built for all determined nutrients using reference materials of known concentrations over the expected range of the field samples. A linear relationship between absorbance and standard concentrations was evaluated (Pearson, R > 0.998) before the start of each analysis. All reagents and reference solutions were made starting from ACS Reagent Grade (Merk KGaA, Darmstadt, Germany and its affiliates).

The chlorophyll *a* concentration was determined according to a spectrofluorimetric method [32]. Water samples (200–1000 mL) were filtered through 47 mm Whatman GF/F filters, which were frozen (−20 °C) until laboratory analysis. The pigments were extracted for 24 h at 4 °C, with 90% acetone from the homogenate filter. The samples were centrifugated at 3000 rpm (SIGMA Laborzentrifugen GmbH, Mod. SBS-LS-1000 SLS, Osterode am Harz, Germany), and the measurements of the chlorophyll *a* were performed using a JASCO FP 6500 spectrofluorometer (JASCO, Europe s.r.l., Cremella (Lecco), Italy).

### 2.3. Phytoplankton Communities

The phytoplankton communities were analyzed by following different approaches. In 1996 a taxonomic analysis of the “Utermohl fraction”, which includes all taxa recognizable under the inverted light microscope, was conducted according to the method described in Section 2.3.3 for cell counting. In addition, in 2007–2008, the size fraction composition was also evaluated, and pico-(0.2–2.0 μm), nano-(2.0–20.0 μm) and micro-phytoplankton components (20.0–200.0 μm) were detected using epifluorescence microscopy.

#### 2.3.1. Picophytoplankton (PPP)

Samples (100 mL) were preserved with formaldehyde (2%) and kept at 4 °C until the laboratory analyses were conducted. Counting was performed using a Zeiss Standard Axioplan epifluorescence microscope (magnification: Plan-Neofluar 100× objective and 10× ocular; HBO 100 W lamp; filter sets: BP 450–490 exciter filter, a FT 510 chromatic beam splitter, and an LP 520 barrier filter). Duplicate slides were prepared from each sample by filtering variable volumes of seawater (10–30 mL, depending on the cell concentration) onto 0.2 μm (pore size) Millipore black membranes. A minimum of 200 cells were counted for each filter within at least 20 randomly selected fields to ensure ±15% confidence levels. The cell number was converted into carbon biomass using a factor of 250 fg C cell^−1^ [33].

#### 2.3.2. Nanophytoplankton (NPP)

To estimate NPP abundance and biomass, samples (125 mL) were preserved in glutaraldehyde (1% final concentration) and stored in the darkness at 4 °C until the analyses were conducted. In the laboratory, sub-samples (20–40 mL) were filtered in triplicate onto black-stained 0.8 μm polycarbonate filters (Ø 25 mm, Whatman^®^ Nuclepore™, Merck KgaA, Darmstadt, Germany)), which were positioned on 1.2 μm nitrocellulose backing filters (Whatman^®^ Millipore, Merck KgaA, Darmstadt, Germany). The enumeration was carried out using a Zeiss Standard Axioplan epifluorescence microscope, as described above. Cells were counted in at least 20 randomly selected fields to give ±15% confidence levels [34]. The cell volume was evaluated by assigning simplified geometrical shapes to cells, or, in some cases, a combination of more geometrical shapes, and then applying or combining standard formulae [35,36]. The carbon content was calculated from mean cell biovolumes, following the method of Strathmann [37].

#### 2.3.3. Microphytoplankton (MPP)

Water samples (500 mL) destined for MPP analysis were fixed with Lugol’s iodine solution to a final dilution of 1.0%, stored at 4 °C, and processed within four weeks. Identification and counting were carried out under an inverted microscope (Labovert FS Leitz equipped with phase contrast), equipped with an AXIOCAM Icc 5 digital camera (Carl Zeiss, Oberkochen, Germany), following the Utermöhl method [38]. According to the observed MPP abundances, a variable volume of the sample (50–100 mL) was settled in an Utermöhl chamber. The minimum value of the counted cells was 200 cells per sample for a confidence limit of 14% [39]. Microalgal cell sizes were measured using the AXIOCAM Icc 5 digital camera (Carl Zeiss, Oberkochen, Germany). The biovolume was calculated by assigning to each cell one geometrical body, or, in some cases, a combination of more geometrical bodies, and applying standard formulae according to Hillebrand et al. [40]. The obtained biovolumes were converted to carbon content using the conversion factors introduced by Menden-Deuer and Lessard [41].

### 2.4. Data Analyses

Data analysis was performed by comparing the two considered sampling periods: before (from January to December 1996) and after the opening of the communication channel of the lagoon with the sea (from April 2007 to March 2008). This comparison was carried out only for the common parameters that were analyzed in both periods.

A principal component analysis (PCA) was applied to characterize and differentiate the two sampling periods on the basis of the environmental features. Preliminarily, the Pearson correlation index between all variables was calculated. Only correlated variables with significant Pearson coefficients (R > 0.3 or R < −0.3) were considered. Moreover, differences in the abundance and biomass values among stations and time (months) were evaluated for environmental and phytoplankton components through a one-way analysis of variance (ANOVA). When significant differences for the main effect were detected (*p* < 0.05), a Tukey’s pairwise comparison test was also applied. Multivariate statistical analysis was used to evaluate the differences in the planktonic community structure between the sampling stations and time. Bi-dimensional representations of the statistical comparisons were obtained by non-parametric multidimensional scaling (nMDS) performed on Bray–Curtis similarity matrices (log-transformed data) [42]. To evaluate the differences in the phytoplankton community assemblages between different stations and times, a one-way analysis of similarities (ANOSIM) was applied. In addition, a one-way similarity percentage procedure (SIMPER routine) was used to obtain the percentage contribution of each phytoplankton taxon to the Bray–Curtis similarity between the groups of samples. The analyses were performed using Primer-E Software package v.7.0 (Plymouth Marine Laboratory, Plymouth, UK), according to Clarke et al. [42], for ANOSIM, SIMPER, and nMDS ordination. The STATISTICA Software package v.10 (StatSoft) was run for the analysis of variance and the correlation analysis.

## 3. Results

### 3.1. Environmental Background

The average values of the environmental parameters collected during the sampling period are shown in Figure 2.

In 1996 and 2007–2008, the water temperature showed a seasonal trend with significant time-related variation (*p* < 0.001), while no significant differences were detected among the stations. Values varied across a wide range, reaching their minima in winter (8.51 ± 0.12 °C, December 1996; 9.75 ± 0.29 °C, February 2008) and maxima in summer (28.1 ± 0.29 °C, June 1996; 34.10 ± 0.30 °C, June 2007).

In both the sampling years, the salinity displayed similar trends and was characterized by high temporal variability, with values increasing from the winter (20.12 ± 2.07 °C, January 1996; 17.53 ± 1.96 °C, March 2008) to the summer (32.5 ± 0.27, July 1996; 34.10 ± 0.30, June 2007) (ANOVA, *p* < 0.001), even if higher values, with the exception of March 2008, were observed in 2007–08 with respect to the 1996 values. In this year, no significant differences were detected among the stations, while in 2007–2008, a reduction in the saline gradient was detected when proceeding from Station 1 to 2 (*p* < 0.05). Such a reduction, on the one hand, could have been due to the combined effects of the seawater input from the channel of communication (Station 1, 32.46 ± 4.79), and on the other, of the inflows of freshwater from Canale Giammatteo (Station 2, 29.72 ± 5.61). Station 3, instead, showed intermediate values compared to those of the other two stations (30.87 ± 4.91).

Dissolved oxygen displayed significantly different values in the two hydrological conditions (*p* < 10^−4^), and higher values were usually detected after the opening of the canal. This physical variable showed a typical seasonal trend, with lower values in the warm period than in the cold period. The values ranged from their minima in August (5.71 ± 0.76 mg L^−1^, 1996; 5.97 ± 0.82 mg L^−1^, 2007) to maxima in February 1996 (10.78 ± 1.33 mg L^−1^) and April 2007 (8.99 ± 1.16 mg L^−1^). In both periods, only significant time-related differences were monitored (*p* < 0.01).

Concerning nutrients, in 1996, significant variations were observed only on the temporal scale, whereas in 2007–2008, significant differences emerged on the spatial scale as well. Ammonia reached higher values in the spring and summer in both periods, and in 2007–2008, it showed significantly higher values at Station 1 (*p* < 0.05). Nitrite+nitrate and phosphate levels were characterized by significant temporal variability (N-NO_2_^−^ + N-NO_3_^−^, *p* < 0.01; P-PO_4_^3−^, *p* < 0.001), and the highest concentrations were detected in the autumn and winter periods. Moreover, phosphates, after the opening of the canal, displayed significant higher concentrations at Station 1 (*p* < 0.05).

Chlorophyll *a* was recorded with significant time-related variations (*p* < 0.001), with an increase in values in the summer. In 2007–2008, significantly higher concentrations were detected at Station 3.

PCA analysis evidenced the differences among the different study periods, confirming the ANOVA results for salinity and dissolved oxygen (Appendix A). Chlorophyll *a* and P-PO_4_^+^ were the only environmental variables poorly correlated with the others (Pearson, R = n.s.), and they were not considered for PCA. The results showed that the first two components explained 76% of the total variance among the variables. Differences in the two sampling periods were due to temperature, salinity, and N-NH_4_^+^ concentration for the PC1, and to the dissolved oxygen level in the PC2 (Appendix A).

### 3.2. Abundance and Biomass of the Phytoplankton Fractions (2007–2008)

The abundance and biomass values of the PPP, NPP, and MPP fractions, detected at all the stations during the sampling period, are displayed in Table 1.

The total abundances of the three phytoplankton fractions ranged between 5.8 × 10^6^ cells L^−1^ and 1.87 x 10^9^ cells L^−1^, while total biomass ranged between 8.1 and 514.0 μg C L^−1^ (Table 1). The highest values of abundances and biomass were detected in the summer months (July–September 2007), and were mainly due to the PPP fraction.

PPP was the dominant component of the community, representing, on average, 99.2% of the total abundance and 46.9% of the total biomass, while NPP (0.7% and 23.5% of the total abundance and biomass, respectively) and MPP (0.1% and 29.6% of the total abundance and biomass, respectively) represented the minor components of the phytoplankton assemblages (Figure 3).

The PPP cell abundances ranged from 5.68 × 10^6^ to 1.87 × 10^9^ cells L^−1^, while those of biomass ranged from 1.42 to 466.70 μg C L^−1^ (Figure 4a). Abundance and biomass showed similar seasonal trends, with marked temporal variations (ANOVA, in both cases, *p* < 10^−4^). The values gradually increased from May to September, reaching the maxima in July and August, while in the resting period, PPP abundance and biomass were negligible. Statistical analyses did not reveal significant differences in the abundance and biomass among the stations, but the highest values were detected at Station 3. The PPP community structure was characterized by the dominance of unicellular coccoid cyanobacteria belonging to the genus *Synechococcus/Cyanobium*. The presence of other picophytoplankton cells was negligible.

NPP abundances ranged from 1.07 to 8.63 × 10^5^ cells L^−1^ (Figure 4b), with a mean value of 2.85 ± 1.77 × 10^5^ cells L^−1^. Biomass ranged from 3.02 to 26.27 μC L^−1^, with a mean value of 8.92 ± 5.48 μC L^−1^. On average, the seasonal abundance and biomass trends were similar and higher from May to July, when peaks were detected in all stations. Abundance and biomass decreased from August onward, and remained almost stable until March 2008. Only at Station 3 did values peak again in December 2007 (up to 6.51 × 10^5^ cells L^−1^ and 19.97 μg C L^−1^). Statistically significant temporal (ANOVA, *p* < 10^−4^) differences were recorded. Moreover, NPP biomass showed significant space-related differences, with the highest values registered at Station 3.

MPP occurred with abundances ranging between 2.19 and 109.76 × 10^3^ cells L^−1^, with a mean value of 41.25 ± 29.30 × 10^3^ cells L^−1^. Biomass varied between 0.23 and 44.75 μg C L^−1^ (Figure 4c), with a mean value of 16.90 ± 13.17 μg C L^−1^. Both variables showed significant time-related variations (ANOVA, *p* < 0.001). The average seasonal cycle of MPP abundance and biomass showed peaks in late spring (June 2007), summer (September 2007), and winter (March 2008); significant temporal variations were also detected (ANOVA, *p* < 0.001). The lagoon was characterized by high MPP spatial variability, with significant differences among stations (ANOVA, *p* < 0.05). The highest values of abundance and biomass were monitored at Station 3 (52.72 ± 32.60 × 10^3^ cells L^−1^ and 22.12 ± 15.09 μg C L^−1^) with respect to those observed at the Station 2 (41.60 ± 32.20 × 10^3^ cells L^−1^ and 17.34 ± 13.85 μg C L^−1^) and Station 1 (29.43 ± 18.31 × 10^3^ cells L^−1^ and 11.25 ± 8.23 μg C L^−1^).

### 3.3. Phytoplankton Community Structure

In 1996, the cell abundances of the phytoplankton Utermöhl fraction ranged from 1.27 to 21.22 × 10^5^ cells L^−1^, with an average value of 9.07 ± 5.28 × 10^5^ cells L^−1^ (Figure 5a). The total biomass varied between 20.9 and 350.9 μg C L^−1^ (average value: 141.4 ± 89.8 μg C L^−1^) (Figure 5b). The highest values were registered in spring and in September and November, but ANOVA did not evidence significant differences in temporal nor in spatial terms.

In 2007–2008, the Utermöhl fraction abundances ranged from 53.6 to 615.2 × 10^3^ cells L^−1^, with a mean value of 179.5 ± 133.5 × 10^3^ cells L^−1^ (Figure 6a). Biomass ranged from 3.2 to 96.4 μg C L^−1^ (average value: 26.2 ± 19.5 μg C L^−1^) (Figure 6b). In this period, phytoplankton showed marked temporal variations, as demonstrated by ANOVA (total abundance: *p* < 0.01; total biomass: *p* < 0.001).

In 1996, 45 taxa were identified (24 diatoms, 18 dinoflagellates, and 3 taxa classified in the “other phytoplankton” group) (Appendix A). Coccolithophores were not observed. In 2007–2008, an increase in the species number was observed, as a total of 58 taxa, including 31 diatoms, 23 dinoflagellates, 1 coccolithophore, 3 species classified in the “other phytoplankton” group, and 3 prokaryotic types were identified (Appendix A Appendix A).

No phytoplankton groups showed significant differences among stations in either period (ANOVA, n.s.). On the contrary, the abundance and biomass of diatoms, dinoflagellates, and “other phytoplankton” displayed significantly different values over time (ANOVA, *p* < 0.05).

The qualitative analysis of the phytoplankton showed that the community was dominated by the “other phytoplankton” in terms of total abundance (1996: 46.3 ± 31.0%; 2007–2008: 56.1 ± 24.7%), while their contribution to the total biomass was lower (1996: 12.5 ± 13.9%; 2007–2008: 25.6 ± 27.1%) (Figure 5). The highest values of abundance (ANOVA, 1996: *p* < 0.05; 2007–2008: *p* < 10^−4^) and biomass (1996: *p* < 0.05; 2007–2008: *p* < 10^−4^) were found in the autumn, winter, and early spring (April) periods. The most conspicuous component was represented by undetermined phytoflagellates <10 μm, which were dominant throughout the year. They represented 90.0% and 40.1% of the “other phytoplankton” total abundances, and 58.7% and 24.2% of the total biomass, in 1996 and 2007–2008, respectively. Undetermined cryptophyceans were also observed in all seasons of 2007–2008, during which they contributed to 55.3% (total abundances) and 66.6% (total biomass) of the “other phytoplankton” group. Filamentous cyanobacteria (e.g., *Oscillatoria* spp. and *Leptolyngbya* spp.) developed mainly in the summer months of 1996, constituting 6.6% and 22.6% of the total abundance and biomass, respectively. In the same period of 2007, the euglenophyceans *Euglena acusformis, Euglena* sp., *Eutreptia viridis,* and *Eutreptiella marina* were detected, with total abundance and biomass values of 4.6% and 9.2%, respectively. Undetermined chlorophyceans were monitored only in autumn 1996, reaching percentage values of 3.4% and 18.7% of the total abundance and biomass, respectively.

Dinoflagellates contributed, on average, to 30.9 ± 24.0% and 24.5 ± 25.4% of the total abundance in 1996 and 2007–2008, respectively. They accounted for 35.7 ± 25.7% and 37.2 ± 31.4% of the total biomass before and after the opening of the canal, respectively. The highest values were usually reached in the late spring and summer periods (abundances *p* < 0.05; biomass *p* < 0.01). The most representative species was *Prorocentrum cordatum*, responsible for a bloom in June 2007 and widespread mainly at Stations 1 and 3. In addition, other species belonging to the genus *Prorocentrum* were identified: *P. micans*, *P. compressum, P. triestinum,* and others potentially producing biotoxins, such as the *Alexandrium minutum* group and *Dinophysis sacculus*.

Diatoms contributed less to the autotrophic assemblages than the other groups, as they were observed with percentage abundance values of 22.8 ± 20.1% in 1996 and 19.1 ± 17.2% in 2007–2008 (Figure 6). Their contribution to the total biomass was, on average, 51.8 ± 24.7% in 1996 and 37.1 ± 28.6% in 2007–2008. The highest values were monitored in the summer and late winter (abundances and biomass *p* < 0.01). Diatoms have always shown uniform distribution within the Lagoon, and were represented by the ticopelagic species *Navicula* sp., *Nitzschia* sp., and *Cylindrotheca closterium,* as well as by *Chaetoceros* spp. and *Thalassiosira* sp. In June 2007, a bloom of the genus *Chaetoceros* developed only at Station 1.

Coccolithophorids were detected only during the second sampling period. They were represented by *Emiliania huxleyi*, which was always present at low values; only in November 2007 at Station 1 was a slight increase in the percentage value observed (3.0% of the total community).

Finally, SIMPER analysis revealed that 73% of the cumulative similarity was due to undetermined phytoflagellates; cryptophyceans; the diatoms *Navicula* spp., *Cylindrotheca closterium,* and *Nitzschia* spp.; and the dinoflagellates *Prorocentrum cordatum* and *P. micans*.

By comparing the community assemblages of the two periods, SIMPER analysis revealed that the phytoplankton species composition in 1996 (Before, B) and 2007–2008 (After, A), displayed significant partitioning in the two considered periods (ANOSIM, global R = 0.563, *p* = 0.1%), confirmed by the non-overlapping between the two assemblages in the nMDS plot (useful stress value) (Figure 7).

SIMPER analysis demonstrated an average dissimilarity of 67.80% between the two phytoplankton assemblages. After the opening of the new canal, increases in the number of taxa were monitored. In particular, diatoms (e.g., *Cylindrotheca closterium*, *Nitzschia* spp., *N. longissima*, *Thalassiosira* sp.) and dinoflagellates (e.g., *Alexandrium minutum* group, *Scrippsiella* spp., *Heterocapsa niei*) were responsible for the difference between the two periods; they were identified for the first time in the Aquatina Lagoon together with the cryptophyceans, euglenophyceans, and *Emiliania huxleyi*.

## 4. Discussion

This study provides a contribution to the knowledge of the evolution of phytoplankton communities subject to alteration of the hydrological regimes of the Aquatina Lagoon as a consequence of a physical modification of the ecosystem’s hydro-dynamism. In particular, the dynamics, distribution, and composition of the phototrophic assemblages before and after the improvement of the connection of the lagoon with sea were studied.

We are aware that the variation in phytoplankton communities may be due to other causes, such as the long-term temporal variation of phytoplankton, a modification of the hydrological conditions due to a change in climate (reduction/increase in rain), or a change in the geomorphological condition due to natural modification of the riparian zone surrounding the Aquatina Lagoon. However, we supported the aforementioned hypothesis because the opening of a new channel led to a change in the environmental variables in the Aquatina Lagoon—in particular, to an increase in salinity and dissolved oxygen that effectively explained the variation observed in the phytoplankton community. As far as nutrients are concerned, with the reduction in nitrogenous compounds (ammonia, nitrites + nitrates), an increase in phosphates was observed. With regard to the phytoplankton biomass expressed in terms of chlorophyll *a*, its values increased on average, demonstrating that the change in the hydrological regime also had positive effects on the productivity of the lagoon. Therefore, this research confirms the importance of an efficient connection with the sea in lagoon environmental management, which has already been observed in other similar systems, such as the Lesina Lagoon (Adriatic Sea, Gargano Peninsula) [9].

The improvement of the water renewal appeared to be beneficial for phytoplankton, which showed particularly higher values in the summer with respect to previous years. This was associated with an increase in the number of taxa observed in the lagoon. Presumably, this plankton component may have been favored by the increase in the phosphate availability. These results are in agreement with previous investigations, which established a phosphorus limitation for the primary production and phytoplankton biomass of the system in the years before the connection with the sea was improved [22,43]. Moreover, these studies suggested meso-oligotrophic conditions for the system by considering the rate at which nutrients were renewed in the water column [22,43]. Our data confirmed these findings and demonstrated that the Aquatina Lagoon is still a highly productive ecosystem today, like other transitional Mediterranean systems [4,9,10,11].

The seasonal trend of phytoplankton showed large fluctuations on an annual cycle, driven both by the seasonal rhythm and by the pulse of nutrients (mainly nitrite plus nitrate). The maximum of the total autotrophic biomass was recorded in summer, as has already been described for other temperate transitional systems, such as the Venice Lagoon [10,11], which are characterized by shallow depths and permanently high nutrient levels [44,45]. On the contrary, the phytoplankton dynamics in the Aquatina Lagoon differed from those observed in other Apulian brackish environments. As an example, in the Lesina Lagoon [9] and in the Mar the Mar Piccolo of Taranto [46,47], higher abundances were detected in the winter–early spring period, while in the Varano Lagoon [48], phytoplankton densities were fairly stable throughout the year, with their oscillations being reduced.

The analysis of phytoplankton based on the size fraction composition revealed the dominance of the smaller component (pico-sized microorganisms) of the community in terms of cell abundance and carbon content. Picophytoplankton, and picocyanobacteria in particular, are considered to be important contributors to the total phytoplankton abundance and biomass in many transitional systems [49,50,51], where they represent >25% of the average annual phytoplankton biomass and productivity, with peak values of up to 100% in summer [52,53]. In the Aquatina Lagoon, picocyanobacteria demonstrated the highest annual phytoplankton abundance and biomass during the summer months, similarly to in other Mediterranean lagoons (e.g., Thau Lagoon) [50]. In fact, it is well known that the high temperature and availability of nitrogen compounds exert significant positive effects on the cyanobacterial growth in transitional systems [53].

The other components (nano- and micro-sized microorganisms) contributed less to the total phytoplankton biomass, but they were important components because they allowed for the comparison of the evolution of the phytoplankton assemblages in the two considered hydrological conditions. In fact, the taxonomical analysis of the “Utermöhl fraction” compared phytoplankton assemblages throughout the years. This approach highlighted a reduction in phytoplankton abundance and biomass following the opening of the new channel. These results are apparently in contrast with the chlorophyll *a* data, which showed a different trend with respect to the microscopical analysis. This is the reason why the European Directives (e.g., the Water Framework Directive, WFD; the Marine Strategy Framework Directive, MSFD) also take into account other metrics (phytoplankton cell abundance and diversity) to evaluate the environmental status of an aquatic system, in addition to the chlorophyll *a* concentration [54,55].

The analysis of the “Utermöhl“ phytoplankton abundance, carbon content, and species composition evidenced a variation in the seasonal trend of diatoms and nano-sized phytoflagellates and an increase in the number of the taxa. In the 2007–2008 sampling period, diatoms were detected in the winter, as in the previous period, but also in the summer. Their presence in the summer is typical of temperate lagoons [48,56], characterized by persistently productive and turbulent conditions [2,10,11]. Some genera detected in the Aquatina Lagoon (e.g., *Thalassiosira* and *Chaetoceros*) produce blooms in transitional waters globally [57].

Another important component of the phytoplankton community in the Aquatina Lagoon is the heterogenous group of phytoflagellates, dominated by the undetermined forms with sizes of <10 μm. Their presence and importance has been already described for other transitional systems [58,59,60,61]. Phytoflagellates have always had a broad temporal distribution, but the opening of the new canal affected the dynamics of these components of the community.

Dinoflagellates were always observed in spring and summer in both survey periods; however, some harmful species appeared (the *Alexandrium minutum* group and *Alexandrium* spp.), while others become more abundant (*Prorocentrum cordatum* and *Dinophysis sacculus*) with respect to previous years. Particularly, the latter two species were typical of the transitional systems [62,63]; *P. cordatum* can give rise to blooms [9] like those in the Aquatina Lagoon. Taking into consideration the high value services (diversity, tourism, and aquaculture) of the Acquatina Lagoon, the presence and dynamics of harmful phytoplankton species should be monitored.

However, it was the analysis and comparison of the assemblage composition that showed the most evident differences in the two examined periods. Considering that the phytoplankton pattern of transitional systems depends on the level of influence of the continental and/or marine inputs, in the Aquatina Lagoon, this effect was evident. In fact, rapid quantitative changes in planktonic communities were observed. Such variations are typical of unstable environments, where different environmental forcings influence the dynamics of communities. In fact, rapid changes in the communities and localized blooms were found predominantly in some stations compared to others. The opening of the canal generally led to an increase in the number of species in the lagoon, favoring the entry of typically marine species (e.g., the diatoms *Chaetoceros* spp. and *Cylindrotheca closterium*; the dinoflagellates of the *Alexandrium minutum* group, *Scrippsiella* sp., and *Heterocapsa* niei; and the coccolithophorid *Emiliania huxleyi*). Statistical analyses demonstrated that the phytoplankton assemblages were significantly different considering the two hydrological conditions.

In conclusion, this research supports the evidence that phytoplankton is a good indicator of environmental status, as its seasonal trends and diversity are closely related to the environmental variables and hydrodynamics of the aquatic systems. This study of the phytoplankton communities in the Aquatina Lagoon contributes to the knowledge of this important component of planktonic communities, which is usually underexplored in many transitional systems (e.g., coastal lagoons). Moreover, the obtained results contribute to the study of the dynamics of coastal lagoons, which is aimed at planning management strategies for the protection and conservation of an ecosystem of high naturalistic value. In particular, with regard to the ecosystem service of the phytoplankton diversity, this research suggests the need to implement long-term monitoring for this essential component of ecosystem functioning by integrating traditional methods (microscopy) with more modern ones (eDNA metabarcoding). This last approach is necessary, above all, to discover the “hidden” diversity of the smaller components, which, especially in recent years, are becoming increasingly important in all aquatic ecosystems.

## Figures and Tables

**Figure 1 microorganisms-11-01277-f001:**
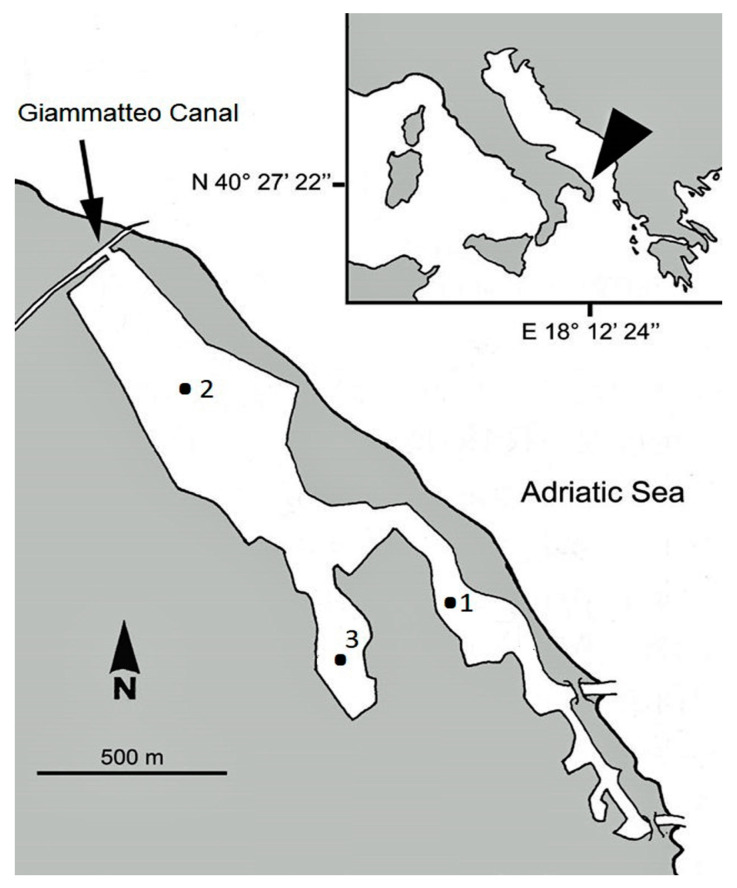
Map of the Aquatina Lagoon, showing the location of the sampling sites.

**Figure 2 microorganisms-11-01277-f002:**
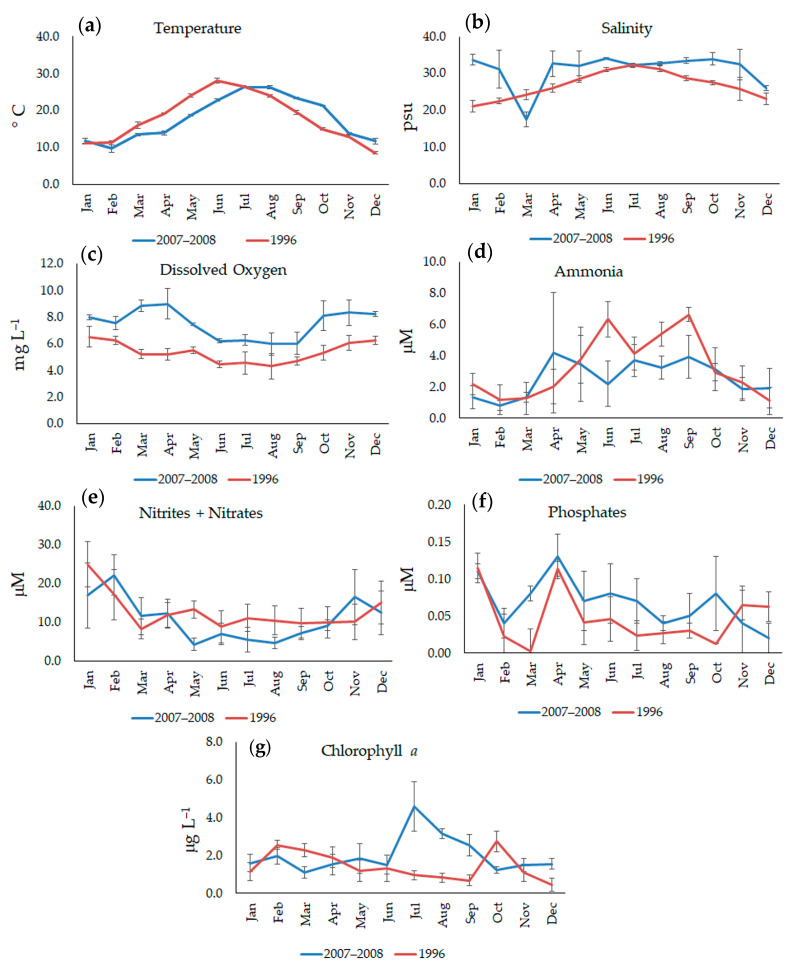
Seasonal trends (average ± SD) of temperature (**a**), salinity (**b**), dissolved oxygen (**c**), ammonia (**d**), nitrites + nitrates (**e**), phosphates (**f**), and chlorophyll *a* (**g**) values observed in the Aquatina Lagoon during the periods of January 1996–December 1996 and April 2007–March 2008. The bars indicate the standard deviation (SD).

**Figure 3 microorganisms-11-01277-f003:**
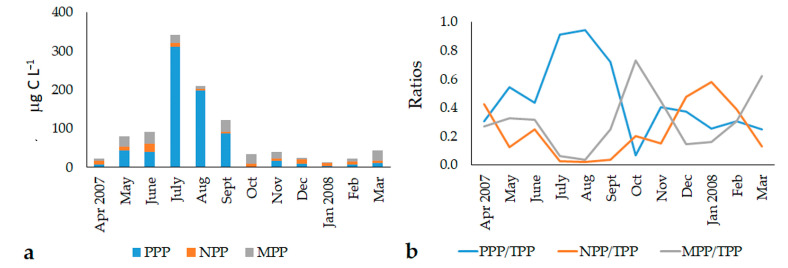
(**a**) Average seasonal trend of the biomass (μg C L^−1^) of phytoplankton fractions in the Aquatina Lagoon during the 2007–2008 samplings; (**b**) picophytoplankton/total phytoplankton Biomass, nanophytoplankton/total phytoplankton biomass, and microphytoplankton/total phytoplankton biomass ratios.

**Figure 4 microorganisms-11-01277-f004:**
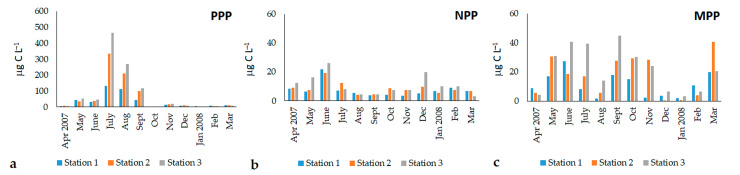
Seasonal trend of the biomass (μg C L^−1^) of phytoplankton fractions in the Aquatina Lagoon during the 2007–2008 samplings in the three sampled stations: (**a**) picophytoplankton; (**b**) nanophytoplankton; (**c**) microphytoplankton.

**Figure 5 microorganisms-11-01277-f005:**
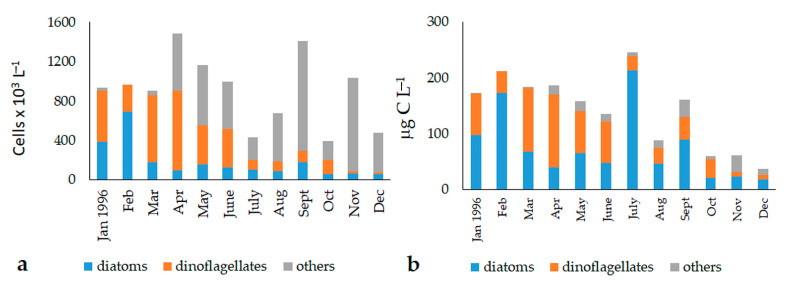
Average monthly values of the “Utermöhl phytoplankton” abundances (**a**) and biomass (**b**) detected in the three stations of the Aquatina Lagoon from January to December 1996.

**Figure 6 microorganisms-11-01277-f006:**
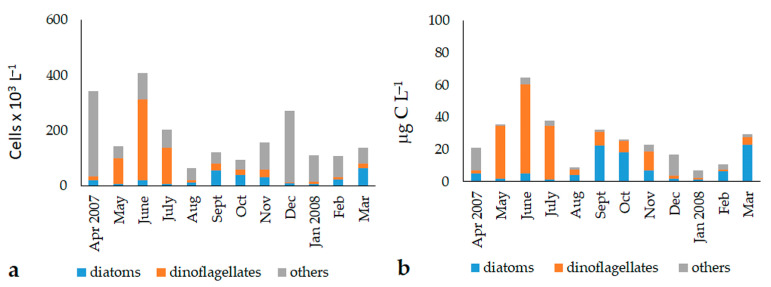
Average monthly values of the “Utermöhl phytoplankton” abundances (**a**) and biomass (**b**) detected in the three stations of the Aquatina Lagoon from April 2007 to March 2008.

**Figure 7 microorganisms-11-01277-f007:**
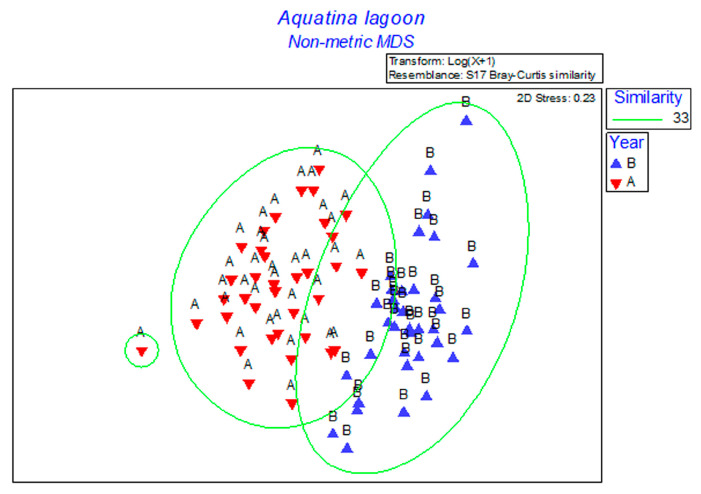
Non-metric multidimensional scaling (nMDS) ordination plot of the phytoplankton species abundances collected in 1996 (B), before the opening of the new canal, and in 2007–2008 (A), after the opening of the new canal. The groups identified by the green line were obtained by overlaying the cluster analysis performed on the same matrix at a similarity level of 33%.

**Table 1 microorganisms-11-01277-t001:** Average ± standard deviation of the abundance and biomass values of the three phytoplankton size components (PPP = pico-; NPP = nano-; MPP = micro-phytoplankton) detected in the Aquatina Lagoon from April 2007 to March 2008.

	PPP		NPP		MPP		Total Fractions	
	Cells × 10^6^ L^−1^	Ug C L^−1^	Cells × 10^3^ L^−1^	Ug C L^−1^	Cells × 10^3^ L^−1^	Ug C L^−1^	Cells × 10^6^ L^−1^	Ug C L^−1^
April 2007	28.59 ± 5.51	7.14 ± 1.38	265.42 ± 55.71	10.04 ± 2.11	22.24 ± 3.18	6.33 ± 2.35	28.88 ± 5.48	23.52 ± 0.69
May	174.28 ± 35.33	43.57 ± 8.83	350.78 ± 195.69	10.03 ± 5.55	32.58 ± 4.35	26.12 ± 7.92	174.66 ± 35.49	79.72 ± 17.67
June	157.67 ± 34.21	39.42 ± 8.55	710.87 ± 180.95	22.51 ± 3.51	76.70 ± 16.83	28.82 ± 11.05	158.46 ± 34.30	90.74 ± 21.34
July	1247.26 ± 668.94	311.81 ± 167.24	310.20 ± 86.40	9.20 ± 2.68	56.89 ± 33.08	21.52 ± 15.88	1247.62 ± 669.00	342.53 ± 182.93
August	791.07 ± 316.68	197.77 ± 79.17	163.83 ± 19.32	4.68 ± 0.56	18.05 ± 16.94	7.23 ± 6.29	791.25 ± 316.68	209.69 ± 50.31
September	351.49 ± 150.31	87.87 ± 37.58	154.20 ± 13.07	4.26 ± 0.26	70.14 ± 29.84	30.13 ± 13.57	351.72 ± 150.35	122.27 ± 50.31
October	9.46 ± 4.55	2.37 ± 1.14	180.33 ± 60.23	6.82 ± 2.28	52.27 ± 15.68	24.90 ± 8.54	9.69 ± 4.61	34.08 ± 11.57
November	66.22 ± 12.88	16.56 ± 3.22	210.06 ± 79.55	6.10 ± 2.32	50.44 ± 31.39	18.17 ± 13.97	66.48 ± 12.98	40.83 ± 19.33
December	36.16 ± 6.47	9.04 ± 1.62	381.91 ± 242.12	11.59 ± 7.63	10.82 ± 7.48	3.55 ± 3.19	36.55 ± 6.59	24.18 ± 10.47
January 2008	13.03 ± 7.89	3.26 ± 1.97	266.47 ± 81.85	7.41 ± 2.31	8.63 ± 4.76	2.05 ± 1.39	13.31 ± 7.88	12.73 ± 4.11
February	27.96 ± 5.66	6.99 ± 1.42	233.65 ± 32.99	8.84 ± 1.25	25.28 ± 5.11	7.02 ± 3.45	28.22 ± 5.65	22.85 ± 5.22
March	42.89 ± 4.55	10.72 ± 1.14	195.76 ± 77.22	5.51 ± 2.15	70.97 ± 33.68	26.98 ± 11.78	43.15 ± 4.63	43.20 ± 13.38

## Data Availability

Not applicable.

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
