# Peer review of "Phytoplankton Size Structure and Diversity in the Transitional System of the Aquatina Lagoon (Southern Adriatic Sea, Mediterranean)"

_microorganisms, 2023, doi:10.3390/microorganisms11051277_

Round 1

Reviewer 1 Report

Review for the paper "Phytoplankton size structure and diversity in the transitional system of Aquatina Lagoon (Southern Adriatic Sea, Mediterranean)" by Carmela Caroppo, Maurizio Pinna, Maria Rosaria Vadrucci submitted to "Microorganisms".

General comment.

In the marine environment, phytoplankton populations exhibit considerable variability on a broad range of timescales, from rapid diel changes in cell size or position in the water column to much longer term and lower frequency interannual oscillations. The variations in plankton diversity and size structure present a particular challenge to marine scientists and resource managers. Not only can they impact biogeochemical cycles, marine food webs, and fisheries, but their causes, links to climate variability, and consequences often remain poorly understood. The authors analyzed the dynamics of phytoplankton communities in the Aquatina Lagoon (Adriatic Sea) focusing on the diversity and size structure of common groups. They revealed a clear seasonal pattern in the abundance and composition of the phytoplankton. Considering the high sensitivity of microplankton to environmental changes, the study provides valuable information that can be used in monitoring research in the region. The manuscript is well structured in all its parts, from the introduction to the discussion. The summary reflects the content of the work but may be slightly improved. The used methodology is appropriate for an ecological paper both for the instrumentation and sampling strategy, with thorough and appropriate statistical data processing. Results are clear and concise, and Discussion contains judgments and comparative data analysis. Tables and figures are enough explicative. There some issues to be addressed by the authors to improve the ms.

Specific remarks.

Abstract. The period of the study must be clearly indicated in the ms. Also, the authors should present data on the diversity and size structure of the phytoplankton.

L24. Consider replacing "in-turn" with "in turn".

L85. Consider replacing "directional" with "temporal".

L96. Consider replacing "conditions" with "systems".

L102. Consider replacing "anticipate" with "work out".

L122-123. Check the units. The indices must be in the upper position.

L131-133 is the repetition of the above text (L125-128) and must be deleted from the ms.

L137-138. In situ must be in Italics.

Fig. 1 must be supplemented with coordinates.

L152. Consider replacing "0.2 *m" with "0.2 µm".

Section 2.6. The authors must carefully describe the procedure to collect and analyze E-DNA samples.

Also, I have found no reference to the methods to determine chlorophyll a although below in the ms the authors provided a graph showing seasonal trend in this parameter (Fig. 2). Include relevant description of the procedure to measure Chl-a.

L199. Consider replacing "comparation" with "comparison".

L208. Consider replacing "extimated" with "applied".

L233. Insert comma after 0C.

Figure 2. The labels (a)-(f) should be added in corresponding plots. The same concern is for Fig. S1.

L299-300. May be 'at all stations’ instead of ‘at each station’?

L401. Consider replacing "The phytoplankton qualitative analysis evidenced a community dominated in terms of abundance by the "other phytoplankton”" with "The phytoplankton qualitative analysis showed that the community was dominated by the "other phytoplankton” in terms of the total abundance".

Section 3.4. The authors should provide more data regarding phytoplankton diversity in the region. Possibly, they may include seasonal comparisons of the diversity and give relevant Figures to visualize the data.

L501-503. Please, provide explanations of the spatial differences in the phytoplankton dynamics between different sites.

L550. Consider replacing "physiognomy" with "pattern".

Discussion. The diversity of the phytoplankton is not discussed at all. Therefore, I encourage the authors to give a section discussing this aspect of the study.

Alternatively, the authors may exclude all mentions regarding E-DNA analysis from the ms without the loss of its quality focusing only on the seasonal cycle of the phytoplankton. However, it is only my opinion that can be omitted by the authors.

Review for the paper "Phytoplankton size structure and diversity in the transitional system of Aquatina Lagoon (Southern Adriatic Sea, Mediterranean)" by Carmela Caroppo, Maurizio Pinna, Maria Rosaria Vadrucci submitted to "Microorganisms".

General comment.

In the marine environment, phytoplankton populations exhibit considerable variability on a broad range of timescales, from rapid diel changes in cell size or position in the water column to much longer term and lower frequency interannual oscillations. The variations in plankton diversity and size structure present a particular challenge to marine scientists and resource managers. Not only can they impact biogeochemical cycles, marine food webs, and fisheries, but their causes, links to climate variability, and consequences often remain poorly understood. The authors analyzed the dynamics of phytoplankton communities in the Aquatina Lagoon (Adriatic Sea) focusing on the diversity and size structure of common groups. They revealed a clear seasonal pattern in the abundance and composition of the phytoplankton. Considering the high sensitivity of microplankton to environmental changes, the study provides valuable information that can be used in monitoring research in the region. The manuscript is well structured in all its parts, from the introduction to the discussion. The summary reflects the content of the work but may be slightly improved. The used methodology is appropriate for an ecological paper both for the instrumentation and sampling strategy, with thorough and appropriate statistical data processing. Results are clear and concise, and Discussion contains judgments and comparative data analysis. Tables and figures are enough explicative. There some issues to be addressed by the authors to improve the ms.

Specific remarks.

Abstract. The period of the study must be clearly indicated in the ms. Also, the authors should present data on the diversity and size structure of the phytoplankton.

L24. Consider replacing "in-turn" with "in turn".

L85. Consider replacing "directional" with "temporal".

L96. Consider replacing "conditions" with "systems".

L102. Consider replacing "anticipate" with "work out".

L122-123. Check the units. The indices must be in the upper position.

L131-133 is the repetition of the above text (L125-128) and must be deleted from the ms.

L137-138. In situ must be in Italics.

Fig. 1 must be supplemented with coordinates.

L152. Consider replacing "0.2 *m" with "0.2 µm".

Section 2.6. The authors must carefully describe the procedure to collect and analyze E-DNA samples.

Also, I have found no reference to the methods to determine chlorophyll a although below in the ms the authors provided a graph showing seasonal trend in this parameter (Fig. 2). Include relevant description of the procedure to measure Chl-a.

L199. Consider replacing "comparation" with "comparison".

L208. Consider replacing "extimated" with "applied".

L233. Insert comma after 0C.

Figure 2. The labels (a)-(f) should be added in corresponding plots. The same concern is for Fig. S1.

L299-300. May be 'at all stations’ instead of ‘at each station’?

L401. Consider replacing "The phytoplankton qualitative analysis evidenced a community dominated in terms of abundance by the "other phytoplankton”" with "The phytoplankton qualitative analysis showed that the community was dominated by the "other phytoplankton” in terms of the total abundance".

Section 3.4. The authors should provide more data regarding phytoplankton diversity in the region. Possibly, they may include seasonal comparisons of the diversity and give relevant Figures to visualize the data.

L501-503. Please, provide explanations of the spatial differences in the phytoplankton dynamics between different sites.

L550. Consider replacing "physiognomy" with "pattern".

Discussion. The diversity of the phytoplankton is not discussed at all. Therefore, I encourage the authors to give a section discussing this aspect of the study.

Alternatively, the authors may exclude all mentions regarding E-DNA analysis from the ms without the loss of its quality focusing only on the seasonal cycle of the phytoplankton. However, it is only my opinion that can be omitted by the authors.

Author Response

Reviewer 1

General comment.

In the marine environment, phytoplankton populations exhibit considerable variability on a broad range of timescales, from rapid diel changes in cell size or position in the water column to much longer term and lower frequency interannual oscillations. The variations in plankton diversity and size structure present a particular challenge to marine scientists and resource managers. Not only can they impact biogeochemical cycles, marine food webs, and fisheries, but their causes, links to climate variability, and consequences often remain poorly understood. The authors analyzed the dynamics of phytoplankton communities in the Aquatina Lagoon (Adriatic Sea) focusing on the diversity and size structure of common groups. They revealed a clear seasonal pattern in the abundance and composition of the phytoplankton. Considering the high sensitivity of microplankton to environmental changes, the study provides valuable information that can be used in monitoring research in the region. The manuscript is well structured in all its parts, from the introduction to the discussion. The summary reflects the content of the work but may be slightly improved. The used methodology is appropriate for an ecological paper both for the instrumentation and sampling strategy, with thorough and appropriate statistical data processing. Results are clear and concise, and Discussion contains judgments and comparative data analysis. Tables and figures are enough explicative. There some issues to be addressed by the authors to improve the ms.

The authors thank the reviewer for the appreciation of our work and for the useful suggestions that have improved it

Specific remarks.

Abstract. The period of the study must be clearly indicated in the ms. Also, the authors should present data on the diversity and size structure of the phytoplankton.

R = The requested information has been added

L24. Consider replacing "in-turn" with "in turn".

R = Replaced

L85. Consider replacing "directional" with "temporal".

 R = Replaced

L96. Consider replacing "conditions" with "systems".

 R = Replaced

L102. Consider replacing "anticipate" with "work out".

 R = Replaced

L122-123. Check the units. The indices must be in the upper position.

R = The units have been checked

L131-133 is the repetition of the above text (L125-128) and must be deleted from the ms.

R = The repletion has been deleted

L137-138. In situ must be in Italics.

R = Done

Fig. 1 must be supplemented with coordinates.

R = Fig 1 has been replaced to improve quality and supplemented with coordinates

L152. Consider replacing "0.2 *m" with "0.2 µm".

R = Done

Section 2.6. The authors must carefully describe the procedure to collect and analyze E-DNA samples.

R = The authors preferred to exclude all mentions regarding E-DNA analysis from the ms, according to the reviewer suggestions.

Also, I have found no reference to the methods to determine chlorophyll a although below in the ms the authors provided a graph showing seasonal trend in this parameter (Fig. 2). Include relevant description of the procedure to measure Chl-a.

R = Procedures for determination of chlorophyll a have been added

L199. Consider replacing "comparation" with "comparison".

R = Replaced

L208. Consider replacing "extimated" with "applied".

 R = Replaced

L233. Insert comma after 0C.

R = Done

Figure 2. The labels (a)-(f) should be added in corresponding plots. The same concern is for Fig. S1.

R = Labels were added in Figure 2 while Fig S1 has been replaced by only one plot to improve information.

L299-300. May be 'at all stations’ instead of ‘at each station’?

 R = Replaced

L401. Consider replacing "The phytoplankton qualitative analysis evidenced a community dominated in terms of abundance by the "other phytoplankton”" with "The phytoplankton qualitative analysis showed that the community was dominated by the "other phytoplankton” in terms of the total abundance".

 R = The paragraph has been replaced

Section 3.4. The authors should provide more data regarding phytoplankton diversity in the region. Possibly, they may include seasonal comparisons of the diversity and give relevant Figures to visualize the data.

R = The authors preferred to exclude all mentions regarding E-DNA analysis from the ms, according to the reviewer suggestions.

L501-503. Please, provide explanations of the spatial differences in the phytoplankton dynamics between different sites.

R = Differences of the phytoplankton seasonal trend in the Aquatina Lagoon respect to the other Apulian transitional systems have been provided.

L550. Consider replacing "physiognomy" with "pattern".

 R = Replaced

Discussion. The diversity of the phytoplankton is not discussed at all. Therefore, I encourage the authors to give a section discussing this aspect of the study. 

Alternatively, the authors may exclude all mentions regarding E-DNA analysis from the ms without the loss of its quality focusing only on the seasonal cycle of the phytoplankton. However, it is only my opinion that can be omitted by the authors.

R = The authors preferred to exclude all mentions regarding E-DNA analysis from the ms, according to the reviewer suggestions.

Comments on the Quality of English Language

English language is fine and only some minor editing is required.

R = The language of the manuscript has been revised

Reviewer 2 Report

Manuscript title: Phytoplankton size structure and diversity in the transitional system of Aquatina Lagoon (Southern Adriatic Sea, Mediterranean)

This study analysed the dynamics and diversity of the phytoplankton communities in different hydrological conditions (before and after the opening of the connection canals). This research is interesting because there is not enough data about changes in phytoplankton structure and dynamics in this area. Furthermore, this research can be valuable in understanding the influence of anthropogenic activities and hydrological regimes on phytoplankton in similar environments. The manuscript is well written and I have some minor comments to improve the manuscript:

Abstract: The abstract needs to be improved. Please add some main results of phytoplankton structure and dynamics and the influence of environmental parameters.

There is no need to repeat words from the title in Keywords (Diversity, Phytoplankton, Aquantina Lagoon, Mediterranean Sea).

Introduction: Please explain the abbreviations when they appeared first time in the manuscript (l.59 E-DNA – move abbreviation from l.64).

Material and methods are clearly written.

Results

l.278: Please also add a figure of the concentration of dissolved oxygen in Figure 2.

l.237-240; l.285-288 etc. The results of environmental parameters changes are not visible in Figure 2. So, I suggest adding a new figure (maybe in the supplement?) with changes of environmental parameters in the three sampled stations as is shown for the seasonal trend of phytoplankton biomass (Figure 4).

Correct Table 1 – delete coma from numbers in the table (i.e. PPP – March 2008 (10,.72±1.14), MPP – June 2007, September 2007, etc.).

Please add years in Table 1S (use this way or some other):

Bacillariophyceae

1996

2007-2008

Achnanthes adnata Bory

+

-

Amphora spp.

+

+

Etc….

 In conclusion, the formal presentation of the paper needs a minor revision.

Minor editing of English language required.

Author Response

Reviewer 2

Comments and Suggestions for Authors

Manuscript title: Phytoplankton size structure and diversity in the transitional system of Aquatina Lagoon (Southern Adriatic Sea, Mediterranean)

This study analysed the dynamics and diversity of the phytoplankton communities in different hydrological conditions (before and after the opening of the connection canals). This research is interesting because there is not enough data about changes in phytoplankton structure and dynamics in this area. Furthermore, this research can be valuable in understanding the influence of anthropogenic activities and hydrological regimes on phytoplankton in similar environments. The manuscript is well written and I have some minor comments to improve the manuscript:

The authors thank the reviewer for the appreciation of our work and for the useful suggestions that have improved it

Abstract: The abstract needs to be improved. Please add some main results of phytoplankton structure and dynamics and the influence of environmental parameters.

R = The requested information has been added

There is no need to repeat words from the title in Keywords (Diversity, Phytoplankton, Aquantina Lagoon, Mediterranean Sea).

R = The indicated keywords were deleted

Introduction: Please explain the abbreviations when they appeared first time in the manuscript (l.59 E-DNA – move abbreviation from l.64).

R = The authors preferred to exclude all mentions regarding E-DNA analysis from the ms, according to the reviewer suggestions.

Material and methods are clearly written.

R = In the “Mat and methods” section, new information was added to respond to the other reviewers’ requests

Results

l.278: Please also add a figure of the concentration of dissolved oxygen in Figure 2. 

R = In Figure 2, the figure representing the dissolved oxygen concentration has been added

l.237-240; l.285-288 etc. The results of environmental parameters changes are not visible in Figure 2. So, I suggest adding a new figure (maybe in the supplement?) with changes of environmental parameters in the three sampled stations as is shown for the seasonal trend of phytoplankton biomass (Figure 4).

R = In many cases, as demonstrated by the ANOVA, environmental variables were not significantly different in the three sampled station, so we added the standard deviation to the average values in the new Figure 2.

Correct Table 1 – delete coma from numbers in the table (i.e. PPP – March 2008 (10,.72±1.14), MPP – June 2007, September 2007, etc.).

R = The Table 1 has been corrected

Please add years in Table 1S (use this way or some other): 

R = The Table 1S has been revised, according the reviewer’ indications

Bacillariophyceae

1996

2007-2008

Achnanthes adnata Bory

+

-

Amphora spp.

+

+

Etc….

 In conclusion, the formal presentation of the paper needs a minor revision.

Comments on the Quality of English Language

English language is fine and only some minor editing is required.

R = The language of the manuscript has been revised

Reviewer 3 Report

Manuscript: Phytoplankton size structure and diversity in the transitional system of Aquatina Lagoon (Southern Adriatic Sea, Mediterranean)

General comments: The manuscript highlights the phytoplankton and abiotic factors’ dynamics in a coastal lagoon before and after the opening of a channel with connection to the Adriatic Sea. Overall, the authors described the impact of hydrological changes in the water quality and ecological status of the lagoon, over two periods. However, some writing issues address some inconsistency which must be revised.

Specific comments can be found as follow:

Lines 38 – 41: It is important to mention the influence of tides in the physical-chemical aspects of the coastal lagoons.

Lines 55 – 58: Please, also mention the need on identifying a diversity representative from both marine and freshwater systems.

Lines 64 – 65: Please, check the definition of eDNA, since this not only represent a "free-DNA" somehow released in the environment, but also from whole cells such as microbes.

Lines 70 – 72: Further information on, e.g., the conservation of the lagoon; any impact, trophic state or human activities in the system would be very welcome for its description as a study area.

Lines 72 – 74: Is this information really relevant for the present manuscript?

Lines 95 – 96: Phytoplankton data are provided only for 2007-2008 period and before the opening of the channels.

Line 97: Add eDNA approach.

Line 146: Please, provide de size spectrum of PPP.

Line 157: Please, provide de size spectrum of NPP.

Line 169: Please, provide de size spectrum of MPP.

Line 184: As I understood, you crossed the phytoplankton composition data found by microscopy analysis with the database for 18S rDNA barcode to check if, e.g., eukaryotic phytoplankton diversity could be assessed by eDNA analysis, right? If so, it is not that clear. Please, clarify and justify why does the 16S rDNA was not assessed since cyanobacteria also occurs in these aquatic systems.

Line 235: Data about stations are described, however no graph or table for comparison are presented. It is difficult to visualize these described fluctuations only with mean values with no SD. The graphs could be improved to include this information. 

Line 279: I suggest you to refer to 2007-2008 and 1996 as after and before the channel opening, respectively, to make easier the understanding.

Lines 285 – 286: Please, consider in these results the same previous comment regarding the station data.

Line 309: This table is using exactly the same information displayed as follow, in the Fig. 3. Please, add any information (e.g. statistical results) that justify this table or it could be deleted. In addition, some cell density vs. biomass data for NPP seems inconsistent. I suggest you to revise them.

Line 318: The graphs quality could be improved. In the figure b the Y axis has no legend or it does not make sense when compared to figure a. Is it in a log-form? Please, modify.

Paragraph at the line 323: Before describing the dynamics in cell abundance and biomass of each phytoplankton size spectrum, it would be welcome to describe the phytoplankton dynamics as a whole (total cell abundance and biomass) and then display the contribution of each size spectrum for that dynamics.

Line 361: Please, configure the Y-axis of the Fig.4b with a maximum value 60 to make it easier to compare.

Line 364: Clarify that data presented here comprise 1996 and 2007-2008 periods. Also, which was the cut-off point to determine these phytoplankton groups as the abundant ones, when euglenids, green algae and cryptophytes were also present?

Line 373 – 374: It is unfair the comparison of only a month in 1996 with an annual variation from 2007-2008. Please, make some ponderation to this statement.

Line 381: The quality of the Fig.5 could be improved.

Line 397: The figure can be improved.

Line 401: Please, be more specific.

Line 437: Please, ponder that despite this result, the data set in both periods was not the same. The period 2007-2008 has a temporal variability.

Line 457: This result section could be better explored.

Line 467: Please, replace “morphology” by “hydrology”.

Line 561: Actually, there is nothing new in this conclusion, but you can emphasize that this pattern has also been seen for coastal lagoons (marine transitional systems) that are little explored for phytoplankton studies.

The manuscript need some minor language revision. 

Author Response

Rewiever 3

Comments and Suggestions for Authors

Manuscript: Phytoplankton size structure and diversity in the transitional system of Aquatina Lagoon (Southern Adriatic Sea, Mediterranean)

General comments: The manuscript highlights the phytoplankton and abiotic factors’ dynamics in a coastal lagoon before and after the opening of a channel with connection to the Adriatic Sea. Overall, the authors described the impact of hydrological changes in the water quality and ecological status of the lagoon, over two periods. However, some writing issues address some inconsistency which must be revised.

Specific comments can be found as follow:

Lines 38 – 41: It is important to mention the influence of tides in the physical-chemical aspects of the coastal lagoons.

R = The effect of tides is important for the “tidal lagoons”, but not for Aquatina Lagoon, which is a non-tidal lagoon. “The relatively shallow depth, very limited tidal regime, scarce and unpredictable rainfall, intermittent freshwater inputs and strong evaporation processes ensure the high spatial and temporal heterogeneity of the main lagoon's chemical–physical characteristics” (from Acquatina -Italy, DEIMS, https://deims.org/8e1909ae-afc0-4207-9314-68e234d57405).         

To clarify this aspect, the authors modified the material and methods (Study area): “Aquatina is a non-tidal lagoon and the tidal regime, on annual basis,……: “

Lines 55 – 58: Please, also mention the need on identifying a diversity representative from both marine and freshwater systems.

R = We added the paragraph “Moreover, in these transitional systems, phytoplankton communities are complex, often represented by taxa typical of both the marine and freshwater systems (Bazin et al., 2014)”.

Lines 64 – 65: Please, check the definition of eDNA, since this not only represent a "free-DNA" somehow released in the environment, but also from whole cells such as microbes.

R = The authors preferred to exclude all mentions regarding E-DNA analysis from the ms, according to the reviewer suggestions.

Lines 70 – 72: Further information on, e.g., the conservation of the lagoon; any impact, trophic state or human activities in the system would be very welcome for its description as a study area.

R = The requested information was added to the ms.

Lines 72 – 74: Is this information really relevant for the present manuscript? 

R = The information has been deleted because it is not relevant, as suggested by the referee

Lines 95 – 96: Phytoplankton data are provided only for 2007-2008 period and before the opening of the channels.

R = Thank you for the suggestion. The authors did not change the introduction (lines 95-96), but clarified in material and methods section that phytoplankton data were provided in terms of the “Utermohl fraction” in 1996 and 2007-2008, moreover they described in the 2007-2008 period (after the opening of the channel). Moreover, we changed the number of the paragraphs.

Line 97: Add eDNA approach.

R = The authors preferred to exclude all mentions regarding E-DNA analysis from the ms, according to another reviewer suggestions.

Line 146: Please, provide de size spectrum of PPP.

R = the information has been provided in the new 2.3 paragraph.

Line 157: Please, provide de size spectrum of NPP.

R = the information has been provided in the new 2.3 paragraph.

Line 169: Please, provide de size spectrum of MPP.

R = the information has been provided in the new 2.3 paragraph.

Line 184: As I understood, you crossed the phytoplankton composition data found by microscopy analysis with the database for 18S rDNA barcode to check if, e.g., eukaryotic phytoplankton diversity could be assessed by eDNA analysis, right? If so, it is not that clear. Please, clarify and justify why does the 16S rDNA was not assessed since cyanobacteria also occurs in these aquatic systems.

R = The authors preferred to exclude all mentions regarding E-DNA analysis from the ms, according to another reviewer suggestions.

Line 235: Data about stations are described, however no graph or table for comparison are presented. It is difficult to visualize these described fluctuations only with mean values with no SD. The graphs could be improved to include this information.  

R = Figure 2 has been modified, by adding to the average trend the SD values, as requested

Line 279: I suggest you to refer to 2007-2008 and 1996 as after and before the channel opening, respectively, to make easier the understanding.

R = the sentence “after the opening of the canal” replaced “in 2007-2008”

Lines 285 – 286: Please, consider in these results the same previous comment regarding the station data.

R = Figure 2 has been modified, by adding to the average trend the SD values, as requested

Line 309: This table is using exactly the same information displayed as follow, in the Fig. 3. Please, add any information (e.g. statistical results) that justify this table or it could be deleted. In addition, some cell density vs. biomass data for NPP seems inconsistent. I suggest you to revise them.

R = The authors disagree with the referee, because in the text the range of the values (abundances and biomass) were reported and not the average ± SD, which is shown in Table 1. The statistical results are reported only in cases when the temporal and/or spatial differences were significant. Moreover, the cell density vs. biomass data were checked. The data are correct.

Line 318: The graphs quality could be improved. In the figure b the Y axis has no legend or it does not make sense when compared to figure a. Is it in a log-form? Please, modify.

R = The quality of all the graphs has been improved. In the figure 3b there is no legend because values are pure numbers, as they represent the ratios among Pico-Phytoplankton/Total Phytoplankton Biomass, Nano-Phytoplankton/Total Phytoplankton Biomass and Micro-Phytoplankton/Total Phytoplankton Biomass.

Paragraph at the line 323: Before describing the dynamics in cell abundance and biomass of each phytoplankton size spectrum, it would be welcome to describe the phytoplankton dynamics as a whole (total cell abundance and biomass) and then display the contribution of each size spectrum for that dynamics.

R = We added the following paragraph: “The highest values of abundances and biomass were detected in summer months (July-September 2007) and were mainly due to the PPP fraction.”

Line 361: Please, configure the Y-axis of the Fig.4b with a maximum value 60 to make it easier to compare.

R = The graph has been modified

Line 364: Clarify that data presented here comprise 1996 and 2007-2008 periods. Also, which was the cut-off point to determine these phytoplankton groups as the abundant ones, when euglenids, green algae and cryptophytes were also present?

R = The authors clarified that the presented data were referred to both the study periods. Moreover, the groups considered were diatoms, dinoflagellates and “other phytoplankton” regardless of their abundance.

Line 373 – 374: It is unfair the comparison of only a month in 1996 with an annual variation from 2007-2008. Please, make some ponderation to this statement.

R = I'm very sorry but I don't understand the referee's observation, as the comparisons made are between two different annual cycles (January-December 1996 and April 2007-March 2008)

Line 381: The quality of the Fig.5 could be improved.

R = Done

Line 397: The figure can be improved.

R = Done

Line 401: Please, be more specific.

R = The paragraph has been rewritten to make it clearer: “The phytoplankton qualitative analysis showed that the community was dominated by the "other phytoplankton” in terms of the total abundance

Line 437: Please, ponder that despite this result, the data set in both periods was not the same. The period 2007-2008 has a temporal variability.

R = We compared data collected in the same three stations during two different years (a monthly samplings from January to December 1996 and from April 2007 to March 2008). Figure 7 represent the differences between the phytoplankton assemblages in the different years.

Line 457: This result section could be better explored.

R = The authors preferred to exclude all mentions regarding E-DNA analysis from the ms, according to another reviewer suggestions.

Line 467: Please, replace “morphology” by “hydrology”.

R = Replaced

Line 561: Actually, there is nothing new in this conclusion, but you can emphasize that this pattern has also been seen for coastal lagoons (marine transitional systems) that are little explored for phytoplankton studies.

R = Thank you very much for the suggestion, which has been added to the conclusions

Comments on the Quality of English Language

The manuscript need some minor language revision. 

R = The language of the manuscript has been revised

Reviewer 4 Report

This manuscript is a case study targeting the Aquatina Lagoon; its objective is to evaluate the phytoplankton dynamics and diversity in two hydrological conditions.

Unfortunately, there are several issues to address, hence for an improved version please consider the following:

L.77, 100 – improper citations – include the links in the reference section …

L.139 – 140 – incomplete information regarding the experimental procedures – do add relevant details about sample preparation, methods used for the determination of nutrients’ concentrations, instrumentation (producers, relevant settings, calibrations), reagents, data processing.

#3.1 – very old datasets – add also new ones and discuss differences

#Figure 2 includes data on chlorophyll a, but there is no method describing how determinations were accomplished – add relevant details about the method or remove the reported data & discussions (L.291 – 293, 479-481, 526, 574)

#Figures 3-6 – resolution problems; these can impact the print quality, hence consider replacing this graphs with proper ones

English language is fine and only some minor editing is required.

Author Response

Reviewer 4

Comments and Suggestions for Authors

This manuscript is a case study targeting the Aquatina Lagoon; its objective is to evaluate the phytoplankton dynamics and diversity in two hydrological conditions.

Unfortunately, there are several issues to address, hence for an improved version please consider the following:

L.77, 100 – improper citations – include the links in the reference section …

R = Done

L.139 – 140 – incomplete information regarding the experimental procedures – do add relevant details about sample preparation, methods used for the determination of nutrients’ concentrations, instrumentation (producers, relevant settings, calibrations), reagents, data processing.

R = Information regarding experimental procedures for determination of nutrients were added

#3.1 – very old datasets – add also new ones and discuss differences

R = This study aims to compare the environmental data and those relating to the phytoplankton communities studied before and after the opening of a connection channel with the sea (1996 and 2007-2008). Even if the data are old, they represent different hydrological situations and have a value for a comparison with the current conditions of the lagoon.

#Figure 2 includes data on chlorophyll a, but there is no method describing how determinations were accomplished – add relevant details about the method or remove the reported data & discussions (L.291 – 293, 479-481, 526, 574)

R = Information regarding experimental procedures for determination of chlorophyll a were added

#Figures 3-6 – resolution problems; these can impact the print quality, hence consider replacing this graphs with proper ones

R = All the figures have been replaced with other of better quality (jpg, 300 dpi)

Comments on the Quality of English Language

English language is fine and only some minor editing is required.

R = The language of the manuscript has been revised

Round 2

Reviewer 3 Report

The authors have performed most of the suggested modifications. However, some minor issues still remain. Please, check the following remarks:

- The figures still need to improve in quality (resolution). Also, Y axis in Fig.3 is needs a description. 

- It is still not clear why only diatoms and dinoflagellates were remarked and the other algae were highlighted to as "other phytoplankton" regardless their abundance.

- The cell density (number) in the Figs. 5 and 6 can be described in the Y-axis in scientific notation.

Author Response

Referee 3

The authors have performed most of the suggested modifications. However, some minor issues still remain. Please, check the following remarks:

- The figures still need to improve in quality (resolution). Also, Y axis in Fig.3 is needs a description. 

R = The quality of the figures has been improved and in Fig. 3 the description of the y axis has been added

- It is still not clear why only diatoms and dinoflagellates were remarked and the other algae were highlighted to as "other phytoplankton" regardless their abundance.

R = The contribution, in terms of total percentages of abundances and biomass, of the microalgae included in the “other phytoplankton” group was detailed, by adding also their temporal distribution.

- The cell density (number) in the Figs. 5 and 6 can be described in the Y-axis in scientific notation.

R = In Figs 5 and 6 the indication has been replaced, as requested

Reviewer 4 Report

Despite this version of the manuscript is much improved compared with the previous one, there a still several issues to consider:

-        L.153-154 - incomplete information regarding the experimental procedures – do add relevant  details about:

a) sample preparation (amount of samples, technique(s) used, reagents..)

b) methods used for the determination of nutrients’ concentrations (spectrophotometry, ion chromatography, membrane selective technique…), c) instruments used (producers, relevant settings, calibrations),

d) reagents (quality, provider)

– I already mentioned this requirement in the former review, but you ignored it

- consider these for all the nutrients you reported: -NH4+, N-NO2, N-NO3, P-PO43)

-        L.154, 155, 159, 179, 184, 190, 208, - replace  all instances of “ml” > mL

-        L.303 – 305 – bad justification for the decision related with  variable selection for PCA – the lack of linearity is not a valid argument, consider rephrasing

-        L.374-378 – delete the added paragraph – filler, it is not necessary to write an introduction for the content, you have only to report your data

Minor editing of English language is still required

Author Response

Referee 4

Comments and Suggestions for Authors

Despite this version of the manuscript is much improved compared with the previous one, there a still several issues to consider:

-        L.153-154 - incomplete information regarding the experimental procedures – do add relevant  details about:

  1. a) sample preparation (amount of samples, technique(s) used, reagents..)
  2. b) methods used for the determination of nutrients’ concentrations (spectrophotometry, ion chromatography, membrane selective technique…), c) instruments used (producers, relevant settings, calibrations), 
  3. d) reagents (quality, provider) 

– I already mentioned this requirement in the former review, but you ignored it

- consider these for all the nutrients you reported: -NH4+, N-NO2–, N-NO3–, P-PO43–)

R = The authors are very sorry that they did not respond effectively to the previous request, but hope they did now. The required details of the nutrient analyses were provided.

-        L.154, 155, 159, 179, 184, 190, 208, - replace  all instances of “ml” > mL

R = All the instances were replaced, as requested.

-        L.303 – 305 – bad justification for the decision related with  variable selection for PCA – the lack of linearity is not a valid argument, consider rephrasing

R=It is true, the lack of correlation among variables is the valid argument. For this reason, in paragraph 2.4 we added a paragraph to explain that a preliminary analysis, based on the correlation analysis, was carried out to choose the variables to be included in the PCA. Moreover, the lines 303-305 were also consequently modified.

-        L.374-378 – delete the added paragraph – filler, it is not necessary to write an introduction for the content, you have only to report your data

R = the paragraph was deleted

The English grammar was also checked.
